# Asymmetric Horizontal Differentiation under Advertising in a Cournot Duopoly

**Malcolm Brady** 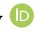

Business School, Dublin City University, D09 Y5N0 Dublin, Ireland; malcolm.brady@dcu.ie

**Abstract:** Horizontal differentiation is generally derived from the aggregate utility function and is assumed to be symmetric. However, empirical work suggests that asymmetric horizontal differentiation can exist in practice. This paper examines the topic of asymmetric horizontal differentiation by allowing a firm's costly advertising to have a different impact on its own demand function than it does on that of its rival. This leads to the interesting analytical result that advertising that increases the cross-price effect of its rival can lead to an increase in firm profits. This introduces the possibility of a 'couple' effect where firm advertising can tilt its own and its rival's demand functions in different directions. Several competitive advertising 'couple' scenarios are explored using numerical simulation.

**Keywords:** differentiation; horizontal; asymmetry; advertising; duopoly; Cournot; simulation





## 1. Introduction

This paper examines a duopoly under competition through asymmetric horizontal differentiation induced by costly advertising. In the literature, horizontal differentiation is generally derived from the aggregate utility function and assumed to be symmetric [1–5]. However, empirical work suggests that asymmetric horizontal differentiation can exist [6]. This paper introduces an asymmetry to horizontal differentiation, allowing a firm's advertising to have a different impact on its own demand function than it does on its rival's demand function. This leads to a novel analytical result regarding asymmetric horizontal differentiation and some interesting new considerations for firm marketers and advertising professionals regarding a 'couple' effect of advertising on horizontal differentiation. These considerations are firstly explored through an analytical model, followed by an examination of four scenarios of 'coupled' advertising using a numerical simulation model based on the analytical model.

## 2. Literature

Bi-modal models of advertising have been proposed in the literature since it was first suggested that advertising can be either informative or persuasive [7] and that the nature of advertising is different for search and experience goods [8,9]. More recently, authors have put forward models of advertising that differentiate between generic advertising, which aims to increase demand for the entire category, and brand advertising, which aims to increase the market share of a specific brand [10–12]. Other authors suggest that firms can engage in mass or targeted advertising [13], or that advertising can be generic where no comparisons are made with competitor products, or comparative where verifiable comparisons are made [14].

The marketing literature tends to model the impact of advertising at the industry level rather than at the firm level and then divide the total market into shares for each firm according to some rules [10–12]. In contrast, the economics literature tends to assume that each firm has its own demand function [1,2]. The model put forward in this paper follows the economics literature in assuming that advertising has an impact on the demand function

of the individual firm rather than on the industry as a whole. It assumes that advertising can impact demand in the following two ways: it can shift the demand function to the right by increasing the reservation price, or it can tilt the demand function by changing its slope [15–17]. Shifting the demand function to the right has the effect of increasing firm demand at every price point and is known as vertical differentiation. Tilting the demand function by altering its slope leads to horizontal differentiation between the products of the two firms. In a competitive situation, more than one firm is involved in the market, and each firm can affect both its own and its rival's demand function. Given that each of the two firms in a competitive duopoly can use advertising to both shift and tilt their own demand functions and also that of their rival through a spillover effect, this leads to a number of different possible advertising scenarios. This paper examines horizontal differentiation, i.e., the tilting of the demand function of the firm and its rival, by altering the slope. In the economics literature, the demand function is derived from the aggregate utility function and is generally assumed to be symmetric [1,2,18], although asymmetric horizontal differentiation has also been theoretically considered [19,20] and found to exist empirically [6]. This paper examines the relatively under-researched area of duopoly behavior when horizontal differentiation is asymmetric.

### 3. Model and Method

The paper builds up an analytical model of differentiated duopoly based on the Cournot–Bowley–Dixit approach [1,2,21,22]. The paper then describes a simulation model based on the analytical model and uses this simulation model to examine the evolution of the industry over time under a number of different competitive scenarios. In particular, the simulations demonstrate the evolution of the Cournot–Nash equilibrium under advertising as firms use advertising to create horizontal diversification in the industry. Simulation is now well recognized as a research approach in the management [23–26] and economic [27] sciences. While simulation does not yield a closed-form solution to the problem, it does provide considerable insight into duopoly behavior under advertising and allows experimental examination of many different scenarios. An alternative way to examine dynamic behavior in duopoly is to use a differential game approach [28–32]; however, while this yields a closed-form solution, it requires the simultaneous solution of two partial differential equations, which is not always feasible and often requires simplifying assumptions such as symmetry.

The basis for the model is the commonly made assumption that price is a decreasing function of the sum of the quantities produced by the two firms [1,2,21,22]. Product differentiation is represented by using the following two different proportionality constants: the own-price effect is the proportionality constant reflecting the impact of own firm quantity on own price, and the cross-price effect is the proportionality constant reflecting the impact of rival firm quantity on own price. The indirect demand functions (Figure 1) for the two firms are formally represented as follows:

$$p_i = a_i - b_i q_i - d_i q_j, \ p_i, q_i, a_i, b_i, d_i \geq 0, \ i = 1, 2, j = 3 - i \tag{1}$$

where $p$ is price, $q$ is the quantity produced, $a$ is the reservation price, $b$ is the own-price effect, $d$ is the cross-price effect, and the subscripts $i$ and $j$ refer to the firm and its rival. Whereas the literature generally assumes that cross-price effect $d$ is symmetric for the two firms [1,2], this paper extends the formulation to allow cross-price effect to be asymmetric. This is represented as $d_1$ and $d_2$ in Equation (1). Authors [19] relate these indirect demand functions to the quasilinear quadratic consumer utility model expressed in matrix form (p. 8, Equations (4) and (4) bis). Authors [6] show that asymmetric horizontal differentiation exists in practice.

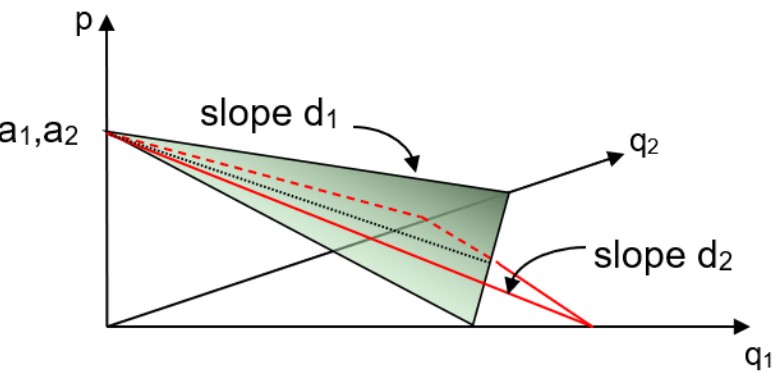

**Figure 1.** Indirect demand function for differentiated duopoly.

Firm profits are represented as follows:

$$\Pi_i = (p_i - c_i)q_i - A_i, \ c_i \geq 0 \ i = 1, 2 \tag{2}$$

where cost is assumed to be a linear function of quantity and $c$ is a proportionality constant representing unit variable cost; $A$ represents expenditure on advertising.

The standard procedure is used to determine the equilibrium point at which the duopoly will operate as follows: substitute expression for price (1) into the profit expression (2); differentiate profit with respect to quantity and set the result to zero to obtain quantity for optimal profit; solve the resulting two equations (usually referred to as reaction functions) to get the following:

$$q_i = \frac{2b_j(a_i - c_i) - d_i(a_j - c_j)}{4b_ib_j - d_id_j}, \ i = 1, 2, j = 3 - i \tag{3}$$

This expression gives quantity for the two firms at the Cournot–Nash equilibrium for an asymmetric duopoly.

Substituting expression (3) for equilibrium quantity into (1) gives equilibrium price as follows:

$$p_i = \frac{2a_ib_ib_j - a_jb_id_i + b_i(2b_jc_i + c_jd_i) - c_id_id_j}{4b_ib_j - d_id_j}, \ i = 1, 2, j = 3 - i \tag{4}$$

Substituting the above expressions for equilibrium quantity (3) and price (4) into (2) gives equilibrium profit as follows:

$$\Pi_i = \frac{\begin{matrix} 4a_i^2b_ib_j^2 - 4a_ib_ib_j(a_jd_i + 2b_jc_i - c_jd_i) + a_j^2b_id_i^2 + 2a_jb_id_i(2b_jc_i - c_jd_i) \\ + b_i(4b_j^2c_i^2 - 4b_jc_ic_jd_i + c_j^2d_i^2) \end{matrix}}{(4b_ib_j - d_id_j)^2} - A_i, i = 1, 2, j = 3 - i \tag{5}$$

Dixit points out that there are two aspects to differentiation, and this affects competition in different ways [1]. Firstly, he suggests that products are differentiated if absolute advantage, $\theta_i = a_i - c_i$, is different for the two firms. Advertising that acts to increase the size of the market by shifting the demand function to the right does so by increasing the value of the reservation price, $a$, thereby creating an absolute advantage for the firm. Advertising that increases the size of the market can also create a competitive advantage for the firm if advertising effectiveness is asymmetric for the two firms. This view is different from that of authors who suggest that advertising that increases demand—generic- or category-building advertising—is cooperative rather than competitive [12].

The exception to this is where products are pure commodities. Here $d = b$ and the values of parameters $a$ and $b$ remain symmetric for both firms for all time; this implies a market where the demand functions of both firms coincide for all time. Advertising shifts this joint demand function to the right; demand for both firms grows but neither firm gains a competitive advantage over the other. In this situation a competitive advantage can

arise only through a cost advantage. Moreover, as both firms gain equally from a firm's advertising there is no incentive for a firm to advertise unless the rival also advertises, and to an equal extent. Unilateral advertising leads to a competitive disadvantage as the non-advertiser gains from advertising-induced industry growth but bears none of the cost. Such a situation suggests that in commodity markets advertising will only take place at industry level and this is borne out in practice where advertising for undifferentiated products such as olive oil, milk, and fruit is often carried out by industry, regional, or national bodies rather than by individual firms.

The second aspect of differentiation suggested by Dixit is known as horizontal and occurs when own-price effect and cross-price effect are different, i.e., $d \neq b$; he formally defines this type of differentiation as $\gamma = \frac{d}{b}$. When products are differentiated then $0 < \gamma < 1$. Note that when the products of the two firms are commodities then $d = b$ and $\gamma = 1$, and when products are totally differentiated, making the firms effectively into monopolies, $d = 0$ and $\gamma = 0$. Firm advertising that tilts the demand function decreases the value of the cross-price parameter $d$. This type of advertising therefore increases the level of differentiation between the products of the two firms. Decreasing the value of the cross-price effect decreases the impact of competition and makes the firm more of a monopolist. At the limit advertising can drive the cross-price effect to zero making the firm a complete monopolist. Although in theory, cross-price effect can take a negative value, implying that products are complements rather than substitutes, this paper does not consider complements and therefore does not allow the cross-price effect to become negative.

Whereas the literature to date has assumed that the cross-price effect is symmetric [1,2] for the two firms the major contribution of this paper is to allow cross-price effect to be asymmetric, i.e., each of the two firms has a different cross-price effect as shown in Equation (1) above. A symmetric horizontal differentiation parameter suggests, under Cournot competition, that a quantity placed on the market by firm two has the same impact on firm one's price as has the same quantity of firm one's product on firm two. This paper takes the view that these impacts can be asymmetric e.g., a quantity of Coca-Cola's product placed on the market may have a different impact on Pepsi's price as will an identical quantity of Pepsi cola's product on Coca-Cola's price. Such asymmetry provides an opportunity to examine more precisely the impact of a change in horizontal differentiation on firm profitability.

To carry out this examination three situations are considered analytically. First, the impact of a change in the firm's own cross-price effect is considered. It can be shown that a decrease in the firm's own cross-price effect will lead to an increase in firm profits. Second, a decrease in the rival firm's cross-price effect will lead to a decrease in firm profits and the converse is as follows: an increase in the rival firm's cross-price effect will lead to an increase in firm profits. Third, if the cross-price effect parameters are the same for both firms then a decrease in cross-price effect will lead to an increase in firm profit. These three effects are now formalized as propositions.

*3.1. Proposition One*

'A Decrease in the Firm's Own Cross-Price Effect Leads to an Increase in Firm Profits'

Differentiating own profit (5) with respect to own cross-price effect gives the following:

$$\frac{d\Pi_i}{dd_i} = \frac{4b_ib_j(2b_j(a_i-c_i)-d_i(a_j-c_j))(d_j(a_i-c_i)-2b_i(a_j-c_j))}{(4b_ib_j-d_id_j)^3}, \qquad (6)$$
$$i = 1,2, j = 3-i$$

When $a_i > c_i$ and $b_i > d_i$, which is true for any feasible duopoly, the first two terms in the numerator and the denominator are positive; when absolute advantage $(\theta_i = a_i - c_i)$ is not too far apart for the two firms then the third term in the numerator is negative making the whole expression negative. Therefore, as own firm cross-price effect decreases, i.e., as

the firm moves towards being a monopolist, firm profits increase. While this is as expected intuitively it is also a novel result.

### 3.2. Proposition Two

'A Decrease in the Rival Firm's Cross-Price Effect Leads to a Decrease in Firm Profits. By Corollary an Increase in the Rival Firm's Cross-Price Effect Leads to an Increase in Firm Profits'

Differentiating own firm profits with respect to rival firm cross-price effect gives the following:

$$\frac{d\Pi_i}{dd_j} = \frac{2b_i d_i (4a_i^2 b_j^2 - 4a_i b_j (a_j d_i + 2b_j c_i - c_j d_i) + (a_j d_i + 2b_j c_i - c_j d_i)^2)}{(4b_i b_j - d_i d_j)^3}$$
$$i = 1, 2, j = 3 - i$$

After multiplying out, collecting terms and factorizing, this can be rewritten as follows:

$$\frac{d\Pi_i}{dd_j} = \frac{2b_i d_i (d_i \theta_j - 2b_j \theta_i)^2}{(4b_i b_j - d_i d_j)^3}, \ i = 1, 2, j = 3 - i \tag{7}$$

where $\theta_i = a_i - c_i$ represents absolute advantage. For $b_i > 0$, $d_i > 0$, and $b_i > d_i$, which are always so by definition, the above expression is always positive. This means that as rival firm's cross-price effect increases (i.e., as the rival firm moves away from monopoly) own firm's profits increase. This suggests somewhat counterintuitively that a firm can increase its profits by making its rival more like itself, i.e., it may be of benefit to a firm to use advertising to un-differentiate its rival from itself. This is a novel result and is a contribution of this paper.

### 3.3. Proposition Three

'If the Cross-Price Effect Parameters Have the Same Value for Both Firms Then a Decrease in Cross-Price Effect Leads to an Increase in Firm Profit'

For completeness, the symmetric situation is examined when cross-price effect is the same for both firms i.e., $d_1 = d_2 = d$; as discussed earlier this is the representation commonly used in the literature [1,2]. Differentiating profit with respect to cross-price effect gives the following:

$$\frac{d\Pi_i}{dd} = \frac{2b_i \left\{ 8a_i^2 b_j^2 d - 2a_i b_j (a_j (4b_i b_j + 3d^2) - 4b_i b_j c_j + d(8b_j c_i - 3c_j d)) + a_j^2 d(4b_i b_j + d^2) \atop + 2a_j (4b_i b_j (b_j c_i - c_j d) + d^2 (3b_j c_i - c_j d)) + (4b_i b_j c_j - d(4b_j c_i - c_j d))(c_j d - 2b_j c_i) \right\}}{(4b_i b_j - d^2)^3} \tag{8}$$
$$i = 1, 2, j = 3 - i$$

The sign of this expression cannot be determined with certainty. However, if the firms have symmetric cost and demand functions, i.e., $a_1 = a_2; b_1 = b_2; c_1 = c_2; d_1 = d_2$, the sub-expression within curly bracket reduces to $-(2b - d)^3$ and the full expression, in turn, reduces to the following: $\frac{d\Pi_i}{dd} = -\frac{2b(a-c)^2}{(2b+d)^3}$.

This is negative for $a > c$, which is always true for real firms. This means that for a symmetric standard differentiated duopoly own profit increases as the differentiation parameter decreases, i.e., profit increases as both firms approach their monopoly point. This is as expected intuitively and is in accord with previous results [1,2].

An interesting outcome of the above examination is that there may be merit in firm advertising in such a way as to produce a 'couple' effect, i.e., to decrease its own cross-price effect and at the same time to increase its rival's cross-price effect. Such a couple effect is now examined using a numerical simulation approach with advertising as the mechanism for altering cross-price effect. To create the simulation model, the following two aspects of advertising need to be considered: the response to advertising and the amount of advertising. Advertising can impact demand in the following number of ways: it can increase vertical differentiation by increasing its own reservation price, or by increasing

or reducing its rival's reservation price, or it can increase horizontal differentiation by reducing its own cross-price effect or by increasing or reducing its rivals cross-price effect.

Vertical differentiation is modeled as a shift in the demand function. Advertising is assumed to have a linear relationship with reservation price as follows:

$$a_i = a_{i_{old}} + \varphi_i A_i + \rho \varphi_j A_j, \ i = 1, 2, j = 3 - i \tag{9}$$

where $\varphi_i$ represents the effect of firms' advertising on their own reservation prices, $A_i$ is the amount (in monetary units) of advertising undertaken by firm $i$, $\rho$ represents advertising spillover, and $A_j$ represents advertising undertaken by the rival firm [33]. The advertising spillover factor $\rho$ represents the proportion of firm advertising that impacts on its rival's reservation price as follows: where $\rho = 0$ firm advertising has no impact on its rival; where $\rho = 1$ firm advertising has the same impact on its rivals as it does on its own reservation price. Reservation price is an accumulator and embodies advertising-generated goodwill. It is assumed that there is no organic growth or decay in reservation price, i.e., the only factor that impacts demand is advertising. The impact of such a shift in the demand function has been examined in [16].

To model horizontal differentiation, which can be imagined as a tilt rather than a shift in the demand function, a linear relationship between advertising and cross-price effect, similar to that of Friedman's [33] approach discussed above, is assumed as follows:

$$d_i = d_{i_{old}} - \sigma_{ii} A_i - \sigma_{ji} A_j, \ i = 1, 2, j = 3 - i \tag{10}$$

where $\sigma_{ii}$ and $\sigma_{ji}$ are own cross-advertising effect and cross cross-advertising effect, respectively, and where advertising acts to alter $d$ the cross-price effect parameter. This formulation also allows for advertising spillover, i.e., firm advertising can impact the cross-price effect of its rival and vice versa. Various scenarios and asymmetries can be easily specified. For example, setting $\sigma_{12} = 0$ and $\sigma_{21} = 0$ restricts firm advertising so as to impact on own firm cross-price effect only. Asymmetry can be imposed by setting $\sigma_{11} \neq \sigma_{22}$ or $\sigma_{12} \neq \sigma_{21}$ i.e., the two firms have different impacts on cross-price effects. Figure 2 demonstrates the impact of advertising on demand.

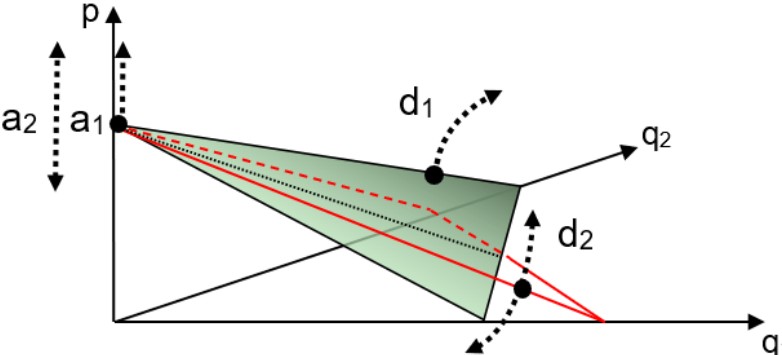

**Figure 2.** Impact of advertising on the indirect demand function.

A specific form of asymmetry provides the following interesting scenario: when $d_1 = d_2 = d$, and $\sigma_{11} = \sigma_{12}$ and $\sigma_{22} = \sigma_{21}$. Here cross-price effects remain identical for both firms for all time even under advertising. This implies that, although firms are differentiated, firm performance will be symmetrical for all time: differentiation may increase profitability of both firms but does not generate a competitive advantage for either firm. This assumption that cross-price effects remain the same for the two firms is the norm in the literature. Another specific asymmetry can be imposed by setting $\sigma_{11} \neq \sigma_{12}$ and $\sigma_{22} \neq \sigma_{21}$ i.e., each firm's advertising affects its own and its rival's cross-price effects differently; under this type of asymmetry advertising can lead to increased profitability and to competitive advantage for the firm.

To determine the amount of advertising the Dorfman–Steiner condition is used [34]. This gives the optimal level of advertising for the firm and is expressed as follows:

$$A = \frac{\eta_A}{\eta} R \tag{11}$$

where $A$ stands for the amount of advertising, $R$ is revenue, $\eta$ is price elasticity of demand, and $\eta_A$ is advertising elasticity of demand. Note that the common rule of thumb used by business managers—where advertising expenditure is taken to be a fixed proportion of revenue—is a specific instance of the Dorfman–Steiner condition where the ratio of elasticities is a constant. Advertising elasticity of demand is the following by definition:

$$\eta_A = \frac{A}{q} \cdot \frac{dq}{dA} \tag{12}$$

To determine $\frac{dq}{dA}$ the demand function (1) is expressed in the following direct form:

$$q_i = \alpha_i - \beta_i p_i - \delta_i p_j, \ p_i, q_i \geq 0, \ i = 1, 2, j = 3 - i$$

where $\alpha_i = \frac{a_i b_j - a_j d_i}{b_i b_j - d_i d_j}$, $\beta_i = \frac{b_j}{b_i b_j - d_i d_j}$ and $\delta_i = \frac{d_i}{b_i b_j - d_i d_j}$.

Expressions for reservation price (9) and cross-price effect (10) are inserted into this expression, and quantity then differentiated with respect to advertising to yield the following:

$$
\frac{dq_i}{dA_i} = \frac{
\begin{aligned}
& a_i b_j (2A_i \sigma_{ii} \sigma_{ij} + A_j(\sigma_{ii}\sigma_{jj} + \sigma_{ij}\sigma_{ji}) - d_i\sigma_{ij} - d_j\sigma_{ii}) \\
& + a_j(A_i^2\sigma_{ii}^2\sigma_{ij} + 2A_i\sigma_{ii}\sigma_{ij}(A_j\sigma_{ji} - d_i) + A_j^2\sigma_{ij}\sigma_{ji}^2 - 2A_jd_i\sigma_{ij}\sigma_{ji} + b_ib_j\sigma_{ii} + d_i^2\sigma_{ij}) \\
& - A_i^2\sigma_{ii}(A_j\sigma_{ii}(\rho\varphi_i\sigma_{jj} - \varphi_j\sigma_{ij}) - b_j\varphi_i\sigma_{ij} - \sigma_{ii}(d_j\rho\varphi_i - p_j\sigma_{ij})) \\
& - 2A_i\sigma_{ii}(A_j^2\sigma_{ji}(\rho\varphi_i\sigma_{jj} - \varphi_j\sigma_{ij}) - A_j(b_j\rho\varphi_j\sigma_{ij} + d_i(\rho\varphi_i\sigma_{jj} - \varphi_j\sigma_{ij}) + \\
& \sigma_{ji}(d_j\rho\varphi_i - p_j\sigma_{ij})) - b_ib_j\rho\varphi_i + b_jp_i\sigma_{ij} + d_i(d_j\rho\varphi_i - p_j\sigma_{ij})) \\
& + A_j^3\sigma_{ji}^2(\varphi_j\sigma_{ij} - \rho\varphi_i\sigma_{jj}) \\
& + A_j^2(b_j(\rho\varphi_j(\sigma_{ii}\sigma_{jj} + \sigma_{ij}\sigma_{ji}) - \varphi_i\sigma_{ji}\sigma_{jj}) + \sigma_{ji}(2d_i(\rho\varphi_i\sigma_{jj} - \varphi_j\sigma_{ij}) + \\
& \sigma_{ji}(d_j\rho\varphi_i - p_j\sigma_{ij}))) \\
& + A_j(b_ib_j(\rho\varphi_i\sigma_{ji} + \varphi_j\sigma_{ii}) - b_j(d_i(\rho\varphi_j\sigma_{ij} - \varphi_i\sigma_{jj}) + d_j(\rho\varphi_j\sigma_{ii} - \varphi_i\sigma_{ji}) + \\
& p_i(\sigma_{ii}\sigma_{jj} + \sigma_{ij}\sigma_{ji})) - d_i(d_i(\rho\varphi_j\sigma_{jj} - \varphi_j\sigma_{ij}) + 2\sigma_{ji}(d_j\rho\varphi_i - p_j\sigma_{ij}))) \\
& + b_ib_j(b_j\varphi_i - d_i\rho\varphi_i - p_j\sigma_{ii}) \\
& + b_j(d_jp_i\sigma_{ii} - d_i(d_j\varphi_i - p_i\sigma_{ij})) \\
& + d_i^2(d_j\rho\varphi_i - p_j\sigma_{ij})
\end{aligned}
}{
\begin{aligned}
& (A_i^2\sigma_{ii}\sigma_{ij} + A_i(A_j(\sigma_{ii}\sigma_{jj} + \sigma_{ij}\sigma_{ji}) - (d_i\sigma_{ij} - d_j\sigma_{ii}) + A_j^2\sigma_{ji}\sigma_{jj} \\
& - A_j(d_i\sigma_{jj} + d_j\sigma_{ji}) - b_ib_j + d_id_j)^2
\end{aligned}
}
$$

$$i = 1, 2; j = 3 - i \tag{13}$$

This expression is inserted into (12) to determine advertising elasticity of demand and, in turn, to determine the optimal amount of advertising using the Dorfman–Steiner condition (11). Note that when advertising impacts only on reservation price and not on cross-price effect (i.e., shifts but does not tilt the demand function) expression (13) reduces to as follows (as used in [16]):

$$\frac{dq_i}{dA_i} = \frac{\varphi_i(b_j - \rho d_i)}{b_ib_j - d_id_j}, \ i = 1, 2; j = 3 - i$$

The simulation model was created by coding up the expressions for price (1), profit (2), quantity (3), and advertising amount (11, 12, 13) using the Powersim simulation software package; the simulation model was then used to examine a number of different competitive scenarios. The model can be fully specified using fifteen parameters representing demand, cost, and advertising effectiveness for the two firms as follows: $a_i, b_i, c_i, d_{i.}, \varphi_i, \sigma_{ij}, i = 1, 2; j = 3 - i$, and $\rho$. Firms are initially symmetric; initial values for

parameters were set at the following: $a_1 = a_2 = 25$, $b_1 = b_2 = b = 0.0001$, $d_1 = d_2 = 0.00005$, and $c_1 = c_2 = c = 8$. These parameters are broadly representative of a fast-moving consumer good in a price-sensitive marketplace. The market demand is such that price decreases by $1 for every 10,000 additional units put on the market by the firm; price also decreases by $1 for every 20,000 additional units put on the market by its rival implying that there exists a considerable level of differentiation between the products of the two firms.

　　Different advertising scenarios may be created by changing the values of the advertising parameters. Parameters *a*, *b*, and *c* remain constant for all simulation runs. As vertical differentiation is not examined in this paper parameters $\varphi$ and $\rho$ are set to zero for all scenarios. The only parameter whose values change during a simulation run is the demand parameter *d*; this parameter is altered as a result of advertising and represents the changing nature of product differentiation over time. Reducing the value of *d* acts to change the nature of the industry from duopoly towards monopoly. The nature of advertising as portrayed in this paper is therefore more persuasive than informative, i.e., advertising is assumed to persuade more people to buy the product or to persuade people that the products are different in nature.

　　Advertising is assumed to act on the cross-price effect and so values for parameters $\sigma_{11}$, $\sigma_{12}$, $\sigma_{21}$, and $\sigma_{22}$ must be set. To see the impact of advertising on demand let parameter $\sigma_{11}$ take a value of 0.0000000002 ($2 \times 10^{-10}$); this means that an advertising expenditure of $0.5 m would decrease the firm's own cross-price effect by 0.00001; i.e., half a million dollars' worth of advertising would decrease the cross-price effect parameter from its initial value of 0.00005 to 0.00004; that in turn would mean that to reduce price by $1 a total of 25,000 rather than 20,000 additional units would have to be put on the market by the rival. In this way advertising has increased the level of product differentiation and reduced the impact of competition in the industry.

## 4. Results

　　Four competitive scenarios were considered. In the first scenario, the cross-price effect remains the same for both firms for all time, i.e., both the firm and its rival advertise so as to tilt upwards both their own and their rival's demand function. By 'tilt upwards' is meant that advertising acts to decrease the value of *d*, the cross-price parameter. The second scenario examines the situation where only one firm advertises so as to tilt both its own and its rival's demand upwards. The third scenario examines the 'couple' effect, where only one firm advertises, but this time it tilts its own demand function upwards and tilts its rival's downwards. The fourth scenario examines the 'couple' effect, where both firms tilt their own demand function upwards and their rival's downwards. For each scenario, the model generates the Cournot–Nash equilibrium values for quantity and the consequent values for price and profitability. Figures 3–6 show the evolution of the Cournot–Nash equilibrium values for the two firms over time for each of the four scenarios.

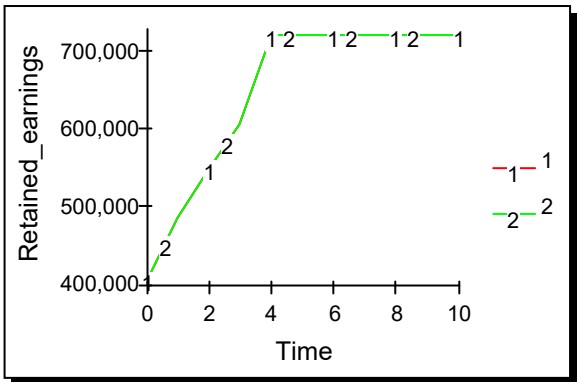

**Figure 3.** Bilateral demand tilting advertising $\sigma_{11} = \sigma_{12} = \sigma_{22} = \sigma_{21} = 1.5 \times 10^{-10}$.

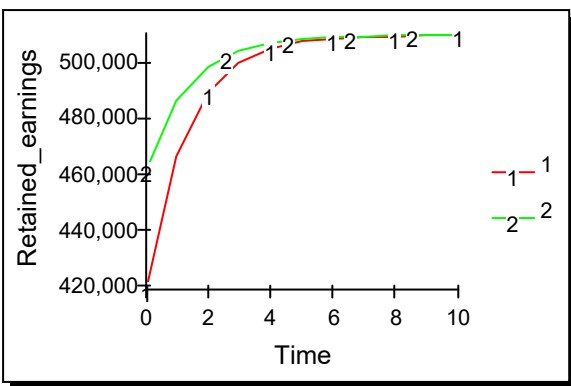

**Figure 4.** Unilateral demand tilting advertising $\sigma_{11} = \sigma_{12} = 1.5 \times 10^{-10}$; $\sigma_{22} = \sigma_{21} = 0$.

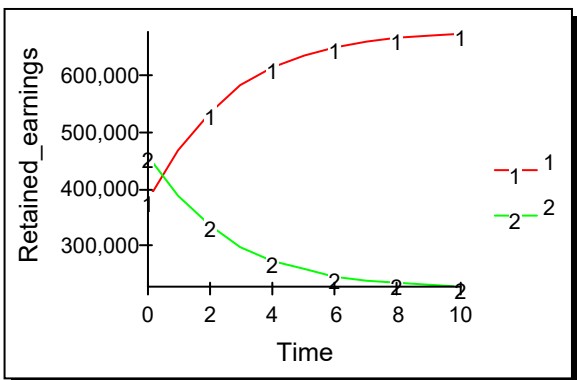

**Figure 5.** Unilateral coupled demand tilting advertising $\sigma_{11} = 1.5 \times 10^{-10}$; $\sigma_{12} = -1.5 \times 10^{-10}$; $\sigma_{22} = \sigma_{21} = 0$.

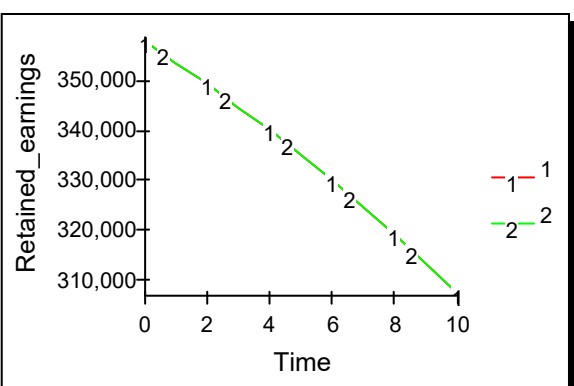

**Figure 6.** Bilateral coupled demand tilting advertising $\sigma_{11} = \sigma_{22} = 1.5 \times 10^{-10}$; $\sigma_{12} = \sigma_{21} = -1.5 \times 10^{-10}$.

### 4.1. Scenario One: Both Firms Advertise So as to Tilt Upwards Both Their Own and Their Rivals Demand Functions

In this situation, the cross-price effect remains the same for both firms for all time, i.e., $d_1 = d_2 = d$, a common assumption in duopoly modeling [1,2]. To achieve this set $\sigma_{11} = \sigma_{22} = \sigma_{12} = \sigma_{21} = \sigma$ to ensure cross-price effects remain symmetric for the two firms. Figure 3 shows the simulation results when $\sigma$ is given a value of $1.5 \times 10^{-10}$. Both firms continue differentiating their products until cross-price effects reach zero in period four. Both firms are now at their monopoly position, advertising is stopped, and consequently, firm growth also stops (recall that the model does not allow the cross-price effect $d$ to become negative; the simulation is programmed to stop further advertising once the cross-price effect reaches zero). Because the firms are symmetric at the beginning of the simulation, and because

both firms are equally effective at advertising, the performance of the two firms remains symmetric. Note that although both firms have improved their performance in an absolute sense, neither firm has gained a competitive advantage over the other.

*4.2. Scenario Two: One Firm Advertises So as to Tilt Upwards Both Its Own and Its Rival's Demand Functions, the Rival Firm Does Not Advertise*

Some form of asymmetry is required for one firm to gain a competitive advantage over the other, for example, in market structure, cost structure, or advertising effectiveness. The effectiveness of advertising that induces horizontal differentiation is now allowed to become asymmetric. Figure 4 shows results when only one firm advertises, but this advertising has an equal impact on its own and on its rival's demand. This is a case of perfectly cooperative spillover as follows: the rival firm gains from the firm's advertising but does not bear any of the cost, consequently gaining a competitive advantage. This result suggests that, as is the case with cooperative shifting advertising, firms will not carry out unilateral cooperative demand tilting advertising as they are in effect providing a public good. However, as both firms lose their absolute advantage as compared with scenario one, there is an incentive for both firms to advertise. There is a prisoners' type dilemma here for firms as follows: it is in their interest to advertise in order to gain an absolute advantage, but it is also in their interest to shirk from advertising in order to gain a competitive advantage.

*4.3. Scenario Three: One Firm Advertises So as to Tilt Its Own Demand Function Upwards and Its Rival's Downwards, Its Rival Does Not Advertise*

This scenario examines the situation where a firm again advertises unilaterally, but in this case, it also acts to tilt downwards the demand curve of its rival. This would make its rival more like itself, i.e., it would be an act of undifferentiation. However, results show that this 'couple' effect—simultaneously tilting its own demand curve upwards and its rival's downwards—yields the firm a significant competitive advantage, although in absolute terms, the firm performs less well than in scenario one (Figure 5).

The results pose the following interesting question for the firm: should it seek the highest absolute performance or the greatest competitive advantage? Seeking the highest absolute performance would encourage the firm to pursue scenario one, i.e., it would advertise in a standard tilting fashion and encourage its rival to do likewise. Seeking to maximize competitive advantage would encourage the firm to pursue scenario three, i.e., it would unilaterally carry out coupled advertising and discourage its rival from advertising. The results point out that maximizing shareholder value (highest absolute performance) does not necessarily result in maximum competitive advantage (highest relative advantage). Some trade-offs of shareholder value against competitive advantage may be required depending on firm priorities. The marketing and strategy literature emphasize market share and competitive advantage, respectively, both of which are relative advantages; the economics and finance literature tends to emphasize maximizing shareholder value, an absolute advantage.

*4.4. Scenario Four: Both Firms Advertise So as to Tilt Own Demand Function Upwards and Rival's Downwards*

This scenario examines the situation when both firms carry out 'coupled' advertising. Both firms now act to tilt upwards their own demand functions and downwards that of their rival. The results show that both firms' performance collapsed (Figure 6). Although the two firms invest in advertising, the two couple effects act against one another and the firms do not move to the monopoly position. However, the cost of advertising lowers the profitability of both firms. Coupled advertising, therefore, provides another prisoner's dilemma situation for firms as follows: if they carry it out unilaterally, they can gain a significant competitive advantage; but if their rival also carries it out, then both are worse off. Seeking to maximize competitive advantage by using coupled advertising would seem to be a risky strategy for a firm as follows: the potential upside gain is large but does

not maximize shareholder value, and the potential downside loss is significant. Seeking to maximize shareholder value using standard advertising is less risky as follows: the potential upside gain is large as the firm achieves monopoly profits, and the downside risk is relatively small, as follows: it may lose some competitive advantage to a rival that shirks.

## 5. Discussion

Advertising has six possible impacts on demand, as shown by the dotted arrows in Figure 2, and given that there are two firms, there are twelve possible firm actions. These actions can also be carried out in combination, and there exist threshold levels and asymmetries, resulting in many possible competitive scenarios, of which this paper examined four. A clear implication from this paper is that the firm needs to be clear about the objective of its advertising—to grow the market or to differentiate the product—as in each case it acts on a different parameter and seeks a different effect. Creating advertising messages to achieve such specific ends may pose an interesting challenge for marketers and strategists.

Under the market share view, generic advertising is seen as category-building and therefore cooperative, while brand advertising is regarded as share-stealing and therefore competitive. This paper has shown that the impact of advertising can simultaneously be both competitive and cooperative. Under demand-tilting advertising, both firms act to horizontally differentiate their own firm products, reducing the impact that the rival product has on firm sales and thereby reducing the impact of competition. In this way, advertising that induces horizontal differentiation can be seen as cooperative. However, if firms are asymmetric in their advertising effectiveness, then tilting advertising alters firm profitability, creating a competitive advantage for one firm over the other. Symmetry is therefore a key factor in determining whether or not the effect of advertising on the firms in a duopoly is competitive or cooperative. Under symmetric conditions, the impact of advertising is mostly cooperative; under asymmetry, the impact may be competitive. Advertising spillover that tilts rival demand upwards (i.e., in the same direction as it tilts its own demand) is cooperative as both firms gain; it is predatory when it tilts rival demand downwards (i.e., in the opposite direction). The paper demonstrates that spillover is rarely beneficial to the firm. When spillover is unilateral, it provides a free ride to the rival, and when it is bilateral, the effects on the firm and rival may act against one another with little benefit gained by either firm. Unilateral predatory advertising is seen to be effective in gaining a competitive advantage over the rival, and also in gaining an absolute advantage; however, it is a risky strategy—if the rival does likewise then both firms decline in profitability.

The existence of asymmetric horizontal differentiation has a number of theoretical implications. In the forward direction, it has implications for firm strategy and marketing. To achieve the 'couple' effect, a firm may need to put out the following two distinct advertising messages: one to people who are primarily its customers and a different message to people who are primarily customers from its rival. The message to its own customers would demonstrate how different its product is to that of its rival in order to further attach them to the firm's product. The message to its rival's customers would demonstrate how close the rival's product is to the firm's product in order to detach them from the rival's product and move them towards the firm's product. Clearly, all this cannot be achieved by a single message, hence the need for two messages, likely via two different media. This 'couple' effect may lead to some interesting new avenues for advertising research.

In the backward direction, it has implications for consumer utility in that a matrix form of quasilinear quadratic utility is required in order to generate asymmetry in horizontal differentiation. The commonly used expression for consumer utility [1,2] will not generate asymmetry because only a single differentiation parameter is available. Martin [20] in his detailed analysis of the origin of linear demand functions examines asymmetry in reservation prices and in quantities per consumer but not for asymmetric differentiation;

extending his analytical approach to include asymmetry in horizontal differentiation may provide an interesting avenue for future research. Matrix approaches to the utility function offer avenues for further research. The elements of the $n \times n$ matrix within the matrix version of the utility function discussed in [19] 'capture the (possibly rich) pattern of complementarity and substitutability among the goods' (p. 7), although the matrix is usually regarded as being symmetric [18]. Authors [35] in their discussion of the demand function do not require the matrix elements to be symmetric, only that the off-diagonal elements are non-negative, the diagonal elements are positive, and the matrix is column diagonally dominant. As asymmetric horizontal differentiation may involve asymmetry in the off-diagonal elements of the matrix, further research into the nature and implications of this matrix may be useful.

Horizontal differentiation has been illustrated as follows: 'Intuitively, if apples are plentiful, consumers are willing to pay less for oranges, all else equal, and vice versa' [20] (p. 7). For the purpose of this paper, the critical clause is 'and vice versa'. If the 'less' that consumers are willing to pay in the vice versa case is exactly the same amount as the 'less' mentioned in the earlier part of the sentence, then the differentiation is symmetric. If the two amounts are different, then horizontal differentiation is asymmetric. In asymmetric horizontal differentiation, the direction of travel makes a difference, i.e., whether you are considering the influence of plentiful apples on oranges or the influence of plentiful oranges on apples. Further theoretical and empirical examination of this direction of travel consideration may provide an interesting area of theoretical and empirical research in economics and marketing.

This research has a number of limitations. Firstly, the model discussed in the paper is theoretical and has not been empirically tested. Secondly, the model is structured so that firms make advertising policy decisions at the beginning and must hold to those policies through time. Learning does not take place, and so firms cannot improve their advertising effectiveness over time. Allowing for improvement of advertising effectiveness over time would introduce an additional feedback loop in the model. This would improve its realism, as in real firms' managers may detect that they are declining vis-a-vis a competitor and may take action to prevent or reduce the rate of decline. Thirdly, the model looks only at tilting advertising; it does not attempt to determine the optimal allocation of advertising between shifting and tilting. This may provide a fruitful avenue for future research. Fourthly, the model assumes that advertising costs are linear. A direction for future research is to explore models that take into account increasing or decreasing returns to advertising. Fifthly, the model assumes that advertising can be precisely targeted as follows: that advertising can be targeted to shift or tilt demand, and that advertising can be targeted to impact or avoid impacting the competitor's demand. It is not clear if in the real world, such precise targeting of advertising can be achieved; however, this may provide an interesting avenue for future marketing research. Finally, while this paper focused on advertising, other means of altering demand exist. For example, product development can also create horizontal differentiation through altering demand parameter $d$. The implications of a 'couple' effect using R&D may provide an interesting avenue for future research.

## 6. Conclusions

This paper examined the relationship between advertising, horizontal product differentiation, firm profitability, and competitive advantage in a duopoly using an analytical model supported by a series of numerical simulation experiments. In contrast to many of the models in the literature, this model allows horizontal differentiation to be asymmetric. The model assumed that advertising tilts the demand function, thereby increasing the level of horizontal differentiation between the two firms, at the limit turning the industry into two separate monopolies. Simulation results show the evolution of the Cournot–Nash equilibrium over time and demonstrate a 'couple' effect whereby advertising can be used to asymmetrically tilt the demand functions of the firm and its competitor, leading to interesting and novel competitive dynamics.

**Funding:** This research received no external funding.

**Institutional Review Board Statement:** Not applicable as study did not involve humans or animals.

**Informed Consent Statement:** Not applicable as study did not involve humans.

**Data Availability Statement:** The study did not report any data.

**Conflicts of Interest:** The author declares no conflict of interest.

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
