# Peer review of "Asymmetric Horizontal Differentiation under Advertising in a Cournot Duopoly"

_games, doi:10.3390/g13030037_

Round 1

Reviewer 1 Report

The authors proposed an interesting analytical and simulation model.
There is no discussion.
The aim of the article has not been clearly stated, there are no hypotheses.
The literature part is quite sparse and requires further development. Few of the most recent literature sources have been used. The literature review should be extracted from the Introduction.

Author Response

The authors proposed an interesting analytical and simulation model.

Response: thank you for this comment and thank you for pointing out areas for improvement. I respond to these individually below.

There is no discussion.

Response: Thank you for this suggestion and for pointing out this omission. I have significantly revised the structure of the paper and I have introduced an extensive discussion section into the paper.

The aim of the article has not been clearly stated, there are no hypotheses.

Response: Thank you for this suggestion. In the extensive revision of the paper and its structure I believe I have now tightened considerably the discussion of the aims and objectives and contribution of the paper and the contribution of the paper. 

The literature part is quite sparse and requires further development. Few of the most recent literature sources have been used.

Response: I have considerably bolstered up the discussion of the literature with the addition of several new and very up to date references. See new references [31-34] in the paper. I have also considerably enhanced the discussion of the micro-foundations of the demand function. 

The literature review should be extracted from the Introduction.

Response: Thank you for this suggestion. I have separated the literature review section from the introduction.

Reviewer 2 Report

Dear Authors,

I red your paper with pleasure. Congratulations. I have some suggestions.

What I miss is a standardized structure. I suggest restructuring your paper as follows: Introduction, theoretical background, methods and data and model, discussion, conclusion.

For scenario 3 it should be explained how a firm can act in details - how a firm can tilt its own demand function upwards and its rival’s downwards by advertising. Perhaps by a form of non cooperative behavior? If so, the theoretical background should contain more from cooperative and non cooperative behavior theory. 

From my point of view the literature review may be deeper. The literature review now seems much concentrated to management science, however behavior simulation is widely used in other fields. See for example https://doi.org/10.3390/data6110109. It may be interesting to include more papers from recent years.

It would make sense to number different scenarios by different chapter numbers. Now all scenarios are numbered 2.1. which may not seem logical in the paper structure.

Good luck!

Author Response

Dear Authors,

I red your paper with pleasure. Congratulations. I have some suggestions.

Response: thank you for this comment and for the suggestions that follow.

What I miss is a standardized structure. I suggest restructuring your paper as follows: Introduction, theoretical background, methods and data and model, discussion, conclusion.

Response: thank you for this suggestion. I have extensively restructured and rewritten the paper along the lines you suggest with the addition of a results section. In doing this however I have endeavored to maintain the essential integrity of the original paper.

For scenario 3 it should be explained how a firm can act in details - how a firm can tilt its own demand function upwards and its rival’s downwards by advertising. Perhaps by a form of non cooperative behavior? If so, the theoretical background should contain more from cooperative and non cooperative behavior theory. 

Response: Thank you for this suggestion. I have given a detailed illustration of how this 'couple' advertising could operated in practice in the discussion section. Thank you also for the suggestion to discuss more and provide more literature in cooperative and non-cooperative behavior. However I think this will be a significant departure from the thrust of the paper and rather than dilute the message of the current paper I would prefer to leave this exploration to a new paper centered on cooperative and non-cooperative behavior in advertising.

From my point of view the literature review may be deeper. The literature review now seems much concentrated to management science, however behavior simulation is widely used in other fields. See for example https://doi.org/10.3390/data6110109. It may be interesting to include more papers from recent years.

Response: Thank you for this suggestion and this reference which I have included. I have also included reference to several additional papers to enhance the theoretical background to the paper. See new references [31-35].

It would make sense to number different scenarios by different chapter numbers. Now all scenarios are numbered 2.1. which may not seem logical in the paper structure.

Response: thank you for pointing this out. This was an oversight and has been rectified along the lines that you suggest.

Good luck!

Response: many thanks for these good wishes!

Reviewer 3 Report

Referee Report on “Asymmetric horizontal differentiation under advertising in a Cournot duopoly”.

Manuscript number games-1678629

This paper examines the topic of asymmetric horizontal differentiation by allowing a firm’s costly advertising to have a different impact on its own demand function than it does on that of its rival.

It is an interesting paper. However, I have the following specific concerns.

Major Concerns and Comments:

  1. The author first uses the individual firm's demand function to explore the impact of product differentiation on firm profits. Second, they introduce the impact of advertising investment on reserve price and product differentiation. Finally, they run a simulation of price, profit, quantity and advertising amount. However, I think the author's model setting is too arbitrary. It is suggested that the authors start from the utility function and deduce how advertising affects consumer reserve price and product differentiation. Please see Ottaviano, Tabuchi, and Thisse (2002), and Hsu et al. (2017).

Reference:

Ottaviano, G. I. P., Tabuchi, T., and Thisse, J. (2002). Agglomeration and trade revisited. International Economic Review, 43, 409–436.

Hsu, C.C., Lee, J.Y. and Wang, L.F.S (2017). Consumer awareness and environmental policy in differentiated mixed oligopoly, International Review of Economics and Finance, 51, 444-454.

  1. The author's cost function for advertising investment is linear, so the impact of advertising investment on profit will not have an interior solution. I suggest that the advertising cost function should be quadratic. Only in this way can an appropriate advertising decision model be constructed, and parameter simulation is meaningful.
  2. It is useful for the author to discuss the influence of advertising investment on the reserve price and product differentiation; however, more important is the influence of the reaction function and equilibrium of the two firms in the duopoly market, and the author is advised to further analyze.
  3. After making revisions, authors should compare the conclusions obtained from their model setting and analysis, and their contribution to the theoretical or practical application of the literature.

Author Response

This paper examines the topic of asymmetric horizontal differentiation by allowing a firm’s costly advertising to have a different impact on its own demand function than it does on that of its rival.

It is an interesting paper. 

Response: thank you for this comment and for raising the specific concerns to which I respond separately below.

However, I have the following specific concerns.

Major Concerns and Comments:

  1. The author first uses the individual firm's demand function to explore the impact of product differentiation on firm profits. Second, they introduce the impact of advertising investment on reserve price and product differentiation. Finally, they run a simulation of price, profit, quantity and advertising amount. However, I think the author's model setting is too arbitrary. It is suggested that the authors start from the utility function and deduce how advertising affects consumer reserve price and product differentiation. Please see Ottaviano, Tabuchi, and Thisse (2002), and Hsu et al. (2017).

Reference:

Ottaviano, G. I. P., Tabuchi, T., and Thisse, J. (2002). Agglomeration and trade revisited. International Economic Review, 43, 409–436.

Hsu, C.C., Lee, J.Y. and Wang, L.F.S (2017). Consumer awareness and environmental policy in differentiated mixed oligopoly, International Review of Economics and Finance, 51, 444-454.

Response: Thank you for providing me with this suggestion and with these two references which have led me to several additional references which I have included in the paper [31-34]. While I accept that many authors begin with the quadratic aggregate utility function this approach invariably leads to a symmetric model of horizontal differentiation because the cross product term has a single differentiation parameter. As I am examining asymmetric horizontal differentiation I must use another approach and so I begin with the firm rather than with the consumer, hence I start with the firms' inverse demand functions. To my knowledge asymmetric horizontal differentiation has been studied relatively little before and this provides the source of the contribution of the paper. I have revised the paper in several places (with changes tracked so that you can follow these revisions) to make the reason for my approach clearer.

2. The author's cost function for advertising investment is linear, so the impact of advertising investment on profit will not have an interior solution. I suggest that the advertising cost function should be quadratic. Only in this way can an appropriate advertising decision model be constructed, and parameter simulation is meaningful.

Response: Thank you for this suggestion. The quadratic cost model that you suggest has some advantages but will also make the analytical elements of the model more complex and in effect produce an entirely new analytical model and consequently a new simulation model and new results. My preference is to remain with the linear cost model for this paper and consider a quadratic cost function in a future paper.  I have placed a note to this effect in the paragraph that discusses limitations and future research directions at the end of the discussion section.

3. It is useful for the author to discuss the influence of advertising investment on the reserve price and product differentiation; however, more important is the influence of the reaction function and equilibrium of the two firms in the duopoly market, and the author is advised to further analyze.

Response: Yes I agree that it is important to discuss the reaction functions and equilibrium in the duopoly. Indeed the simulation model solves the reaction functions and determines the evolution of the Cournot-Nash equilibrium over time as advertising changes the demand functions of the two firms; figures 3 to 6 depict this evolution of equilibrium graphically. I have revised the discussion section to make this point clearer in the paper.

4. After making revisions, authors should compare the conclusions obtained from their model setting and analysis, and their contribution to the theoretical or practical application of the literature.

Response: Thank you for this suggestion. I have extensively revised the discussion section in light of this suggestion and your other suggestions.

Round 2

Reviewer 2 Report

Dear Authors, congratulations for your paper. I consider it suitable

for publication in present form.

Reviewer 3 Report

It's a pity that the setting of advertising costs has not been revised, but I appreciate the author's efforts.